

# Comparative genomic analysis of the principal *Cryptosporidium* species that infect humans

Laura M. Arias-Agudelo[1], Gisela Garcia-Montoya[1], Felipe Cabarcas[1,2], Ana L. Galvan-Diaz[3] and Juan F. Alzate[1]

[1] Centro Nacional de Secuenciación Genómica - CNSG, Sede de Investigación Universitaria - SIU, Departamento de Microbiología y Parasitología, Facultad de Medicina, Universidad de Antioquia, Medellin, Antioquia, Colombia

[2] Grupo SISTEMIC, Departamento de Ingeniería Electrónica, Facultad de Ingeniería, Universidad de Antioquia, Medellin, Antioquia, Colombia

[3] Grupo de Microbiología ambiental. Escuela de Microbiología, Universidad de Antioquia, Medellin, Antioquia, Colombia

Corresponding author
Juan F. Alzate,
jfernando.alzate@udea.edu.co

## ABSTRACT

*Cryptosporidium* parasites are ubiquitous and can infect a broad range of vertebrates and are considered the most frequent protozoa associated with waterborne parasitic outbreaks. The intestine is the target of three of the species most frequently found in humans: *C. hominis*, *C. parvum*, and. *C. meleagridis*. Despite the recent advance in genome sequencing projects for this apicomplexan, a broad genomic comparison including the three species most prevalent in humans have not been published so far. In this work, we downloaded raw NGS data, assembled it under normalized conditions, and compared 23 publicly available genomes of *C. hominis*, *C. parvum*, and *C. meleagridis*. Although few genomes showed highly fragmented assemblies, most of them had less than 500 scaffolds and mean coverage that ranged between 35X and 511X. Synonymous single nucleotide variants were the most common in *C. hominis* and *C. meleagridis*, while in *C. parvum,* they accounted for around 50% of the SNV observed. Furthermore, deleterious nucleotide substitutions common to all three species were more common in genes associated with DNA repair, recombination, and chromosome-associated proteins. Indel events were observed in the 23 studied isolates that spanned up to 500 bases. The highest number of deletions was observed in *C. meleagridis*, followed by *C. hominis*, with more than 60 species-specific deletions found in some isolates of these two species. Although several genes with indel events have been partially annotated, most of them remain to encode uncharacterized proteins.

## INTRODUCTION

*Cryptosporidium* is a ubiquitous enteric apicomplexan that infects a broad range of vertebrates, including humans and domestic and wild animals (*Khan, Shaik & Grigg, 2017*). It is described as an important cause of chronic diarrhea in AIDS and other

immunocompromised patients. It is also a cause of death in children under 24 months old, especially in low-income countries (*Sow et al., 2016*; *Troeger et al., 2017*). Furthermore, *Cryptosporidium* is the most frequent protozoa associated with waterborne parasitic outbreaks worldwide (*Efstratiou, Ongerth & Karanis, 2017*).

*Cryptosporidium* is classified as a gregarine, within its subclass, the Cryptogregaria (*Ryan et al., 2016*). Thus far, there are at least 39 species established, and more than 30 genotypes (*Firoozi et al., 2019*). Among them, approximately twenty-one have been found in humans. However, *Cryptosporidium hominis* and *Cryptosporidium parvum* are responsible for more than 90% of the reported human infection cases worldwide (*Grinberg & Widmer, 2016*; *Feng, Ryan & Xiao, 2018*). Other species, occasionally described in humans, including *C. meleagridis*, *C. felis*, *C. canis*, *C. ubiquitum*, *C. cuniculus*, *C. viatorum*, *C. muris*, chipmunk genotype I, *C. andersoni*, *C. suis*, *C. bovis*, horse genotype, *C. xiaoi*, skunk genotype, mink genotype, *C. erinacei*, *C. fayeri*, *C. scrofarum* and *C. tyzzeri* (*Feng, Ryan & Xiao, 2018*). These species and genotypes differ significantly in human infectivity, host range, geographic distribution, and virulence (*Camaa et al., 2007*; *Cama et al., 2008*; *Adamu et al., 2014*; *Feng, Ryan & Xiao, 2018*). Some species, such as *C. hominis*, have very narrow host ranges, mostly restricted to humans, nonhuman primates, and horses, whereas others, such as *C. parvum*, have a broad host range, infecting ruminants, horses, rodents, and other animals besides humans (*Ryan, Fayer & Xiao, 2014*). *Cryptosporidium meleagridis*, which is the third most prevalent species in humans, has been described in mammals and birds (*Stensvold et al., 2014*).

*Cryptosporidium* molecular phylogeny and evolutionary relationships have been studied through PCR of single or multiple genetic markers (*Feng et al., 2007*; *Xiao, 2010*; *Xiao & Feng, 2017*; *Cunha, Peralta & Peralta, 2019*). *Cryptosporidium* species identification protocols usually include the amplification and sequencing of the small subunit (SSU) rRNA gene. Additionally, glycoprotein 60 gene (gp60) has been used to study subtypes and intra-species diversity of the genus, leading to the currently accepted classification into gp60 allelic families subtypes (*Xiao, 2010*). The SSU rDNA gene in *Cryptosporidium* species does not evolve under a neutral model, and its genetic diversity is restricted to a few polymorphic sites (*Sulaiman, Lal & Xiao, 2002*). Additionally, some species appear to evolve much quicker than others, according to the SSU rDNA locus.

Regarding the gp60 gene, its high genetic diversity has been attributed to the action of positive selective pressure. Because it can be exchanged by genetic recombination, its typing information does not always agree with other loci, especially with some *C. parvum* subtype families (*Feng, Ryan & Xiao, 2018*). Other loci used in the *Cryptosporidium* typing include coding genes of actin, 70 kDa heat-shock protein (HSP70), and the *Cryptosporidium* oocyst wall protein (COWP) (*Cunha, Peralta & Peralta, 2019*). Most of these loci do not evolve neutrally, and the rate of evolution may vary among the different species of the genus (*Sulaiman, Lal & Xiao, 2002*). According to the above, phylogenetic inference based on analysis of one or a few genetic loci, some under selection pressure, might not reflect the real phylogenetic relationships at the whole genome level (*Sulaiman, Lal & Xiao, 2002*; *Morris et al., 2019*).

The growing use of whole-genome shotgun sequencing (WGS) and next-generation sequencing (NGS) in the study of *Cryptosporidium* spp. is allowing better phylogenetic and comparative genomic analysis within the genus (*Widmer & Sullivan, 2012*; *Mazurie et al., 2013a*; *Guo et al., 2015*; *Isaza et al., 2015*; *Ifeonu et al., 2016*; *Beser et al., 2017*; *Sikora et al., 2017a*; *Feng et al., 2017*; *Khan, Shaik & Grigg, 2017*; *Gilchrist et al., 2018*; *Fan, Feng & Xiao, 2019*; *Xu et al., 2019*; *Nader et al., 2019*). The first *Cryptosporidium* genomes were generated in 2004 using capillary sequencing, belonging to the *C. parvum* (Iowa strain) and *C. hominis* (TU502 strain) species (*Xu et al., 2004*; *Abrahamsen, 2004*). Since then, high-quality (NGS-based) genomes from several subtypes of these species and other human-related species are increasingly available (*Widmer & Sullivan, 2012*; *Mazurie et al., 2013a*; *Guo et al., 2015*; *Beser et al., 2017*; *Feng et al., 2017*; *Sikora et al., 2017a*; *Gilchrist et al., 2018*; *Nader et al., 2019*; *Xu et al., 2019*). Comparative analysis shows a remarkable structural and compositional conservation in genome organization among intestinal *Cryptosporidium* species (*Fan, Feng & Xiao, 2019*). The genome size is near 9.0 Mb in length and is arranged into eight chromosomes, with perfect synteny (no evidence of genome rearrangements), extremely compact coding genes, and a low number of gene introns (*Fan, Feng & Xiao, 2019*). It has been postulated that the phenotypic differences between *Cryptosporidium* species could be associated with minor sequence variations (single nucleotide variants-SNVs and short indels) that can affect expressed proteins or gene regulation patterns (*Guo et al., 2015*; *Feng et al., 2017*; *Sikora et al., 2017a*; *Gilchrist et al., 2018*; *Su et al., 2019*; *Xu et al., 2019*). Nonetheless, several studies have demonstrated events of major insertions and deletions between several species of *Cryptosporidium*, usually involving members of multicopy gene families under positive selection, such as those located near telomeres, like the MEDLE proteins, insulinase-like proteases, and mucin-type glycoproteins. These genes have been associated with the host-parasite interaction (*Guo et al., 2015*; *Feng et al., 2017*; *Gilchrist et al., 2018*; *Xu et al., 2019*).

Molecular phylogenetic strategies have been developed to understand the evolutionary relationships between proteins or genes and help to unravel the evolutionary history of the species. Phylogenetic analysis can also give insights into epidemiological, immunological, and evolutionary processes shaping genetic variation in natural populations of the parasite, and may even have the potential to improve future public health measures (*Baele et al., 2017*; *Theys et al., 2019*). Although phylogenomics studies on apicomplexan parasites are scarce, in models such as Piroplasmida and Haemosporida, they have been useful in the elucidation of the taxonomy and phylogenetic relationships within these protozoa, through the incorporation of a broad number of taxons and DNA datasets (*Cornillot et al., 2012*; *Lack, Reichard & Van Den Bussche, 2012*; *Galen et al., 2018*).

Here we used a comprehensive phylogenomic approach to have a better view of the evolutionary relationships of the three most relevant human infecting species of *Cryptosporidium* protozoan parasites (*C. parvum*, *C. hominis*, and *C. meleagridis*), including genomes with different subtype families. Furthermore, a detailed comparative analysis of the largest indel events detected in these three species is presented, which allowed a more comprehensive view of the gene content differences among these three apicomplexan species.

## METHODS

### NGS data and assembly

Read sequences were downloaded from the Sequence Read Archive - SRA of the NCBI public database. Reads of 23 genomic projects of three *Cryptosporidium* species were included: *C. parvum, C. hominis,* and *C. meleagridis*. Four genomes of *C. meleagridis*, ten genomes of *C. hominis,* and nine genomes of *C. parvum*, including the *C. parvum* anthroponotic isolates UKP14 and UKP15 were analyzed. The reads were extended using FLASH v1.2.11 (*Magoc & Salzberg, 2011*); then the extended and independent read pairs were assembled with the software SPADES v3.11.1 (*Bankevich et al., 2012*), with default settings and testing k-mers of 33, 55, 77, 99, and 111 bases; then the descriptive statistics of the assembly were calculated with in house Perl scripts. To avoid any possible contamination of the sequences with other species, the assembled contigs were aligned using BLASTN with the reference genomes (chromosomes) of *C. hominis* UdeA01, *C. parvum* Iowa II and *C. tyzzeri* UGA55, downloaded from CryptoDB v43 (*Puiu, 2004*). The contigs with a Bit score value ≥ 300 were kept for further analyses.

### Sequencing depth analysis

All the reads were mapped against the *C. parvum* Iowa II reference genome downloaded from CryptoDB v43 and against the *de novo* assembled contigs using BWA (*Li & Durbin, 2010*) with default options. Then, the Samtools-depth tool (*Li et al., 2009*) was used to compute the read depth at all positions, with a maximum coverage depth to 1,000,000. Finally, the mean coverage was estimated in the Linux terminal with an awk formula.

### Single nucleotide variants detection

Detection of SNVs was performed aligning with MUMmer v3 the assembled scaffolds for each isolate with the *C. parvum* Iowa II reference genome. Then, the MUMmer function show-SNPs, and dnaDIFF tools were used (*Delcher et al., 2002*).

### Phylogenomic analysis

The phylogenomic analysis was performed based on the SNVs found within the 24 selected *Cryptosporidium* genomes. A matrix comprising all the SNVs detected was constructed and loaded into the program IQ-TREE v1.6.12 (*Trifinopoulos et al., 2016*). A maximum-likelihood (ML) tree (*Felsenstein, 1981*) was built using this phylogenomic inference software. Modelfinder was used to select the best model of evolution under Bayesian Information Criterion (BIC) (*Schwarz, 1978*). Transversion Model TVM (TVM+F+ASC+R2) was selected as an evolutionary model. Branch support was estimated using 1000 iterations of the Shimodaira-Hasegawa test (SH -aLRT) (*Shimodaira & Hasegawa, 1999*),Shimodaira Hasegawa 1000 pseudoreplicates of ultra-fast bootstrap (*Hoang et al., 2018*). The presented tree is unrooted, and the longest branch was selected as an arbitrary outgroup. The tree was edited using FigTree v1.4.4 (*Andrew, 2018*).

### Identification of SNVs in coding regions

For the detection of single nucleotide variants in the CDSs, the reads were mapped to the genome of *C. parvum* Iowa II with the BWA (Burrows-Wheeler Aligner) aligner version

0.7.17 (*Li & Durbin, 2010*). Later, the bam and VCF files were generated with SAMtools and BCFtools, respectively (*Danecek et al., 2011*). Only variants with a Phred quality score ≥ 50 (*Holder et al., 2013*) were included. Finally, the SAMtools Depth tool was used to calculate the reading depth at all positions, with a maximum coverage depth of 1,000,000. Subsequently, the mean coverage was estimated with an awk formula. The single nucleotide variants in CDSs were annotated with the effect predictor SIFT4G (sorting intolerant from tolerant) version 3.0 (*Vaser et al., 2016*).

## Identification of insertions and deletions

Insertions and deletions –indels - were identified with MUMmer aligner (Maximal Unique Matches) version 3.0 (*Delcher et al., 2002*), as previously described by *Isaza et al. (2015)*, excluding those with a length <50 nucleotides with the software online Assemblytics (*Nattestad & Schatz, 2016*). To perform the functional analyzes, the genome of *C. parvum* Iowa II version 43 deposited in CryptoDB (*Puiu, 2004*) was used as a reference. Variants were initially detected in coding regions, and then genes with shared and species-exclusive deleterious mutations were identified and annotated.

## Identification and annotation of genes with deleterious non-synonymous changes

From the prediction obtained with SIFT4G, genes with deleterious mutations in the eight chromosomes were filtered for each of the genomes. These genes were extracted with the SeqSelect.py script, and enrichment analysis in Gene Ontology terms was performed with the EggNOG mapper program version 1.0 (*Huerta-Cepas et al., 2017*); and visualization with WEGO (Web Gene Ontology Annotation Plotting) version 2.0 was done (*Ye et al., 2018*). Then, functional orthology analysis was performed with KEGG on the KAAS server (*Ogata et al., 1999*), selecting the best hits using a bidirectional strategy - BBH (bi-directional best hit). To detect genes with transmembrane domains, the TMHMM version 2.0 server (*Krogh et al., 2001*) was used. Prediction of the genes that code for proteins with classical and non-classical secretion was performed with the online servers SIGNALP version 5.0 (*Almagro Armenteros et al., 2019*) and SECRETOME-P version 2.0 (*Bendtsen et al., 2004*), respectively.

# RESULTS

## Selected *Cryptosporidium* genomes

To carry out a comprehensive genomic comparative analysis, we chose all the NGS projects of the three main species of *Cryptosporidium* that infect humans (C. *hominis, C. parvum*, and *C. meleagridis*), that have raw sequence data available in public databases. A total of 23 genomes were included: ten isolates of *Cryptosporidium hominis* (UKH1, UKH3, UKH4, UKH5, 37999, TU502_2012, 30976, UdeA01, SWEH2, and SWEH5); nine isolates of *C. parvum* (UKP2, UKP3, UKP4, UKP5, UKP6, UKP7, UKP8, and the *C. parvum* anthroponotic isolates UKP14 and UKP15); and four isolates of *C. meleagridis* (UKMEL1, UKMEL3, UKMEL4, and TU1867). The low representation of *C. meleagridis* genomes is due to the limited number of sequencing projects for this species, so all available genomes

in public databases until September 2019 were included. The genomes belong to different gp60 subtypes, and the majority comes from the UK. All the studied genomes come from parasite oocysts isolated from human feces of patients with natural infections, and four of these were maintained through passages in piglets, mice, or chickens. Most genomes were sequenced on Illumina's HiSeq and MiSeq platforms, and only two (*C. hominis* SWEH2 and SWEH5) used the Ion Torrent platform (Table S1).

### *De novo* genome assembly analysis

The raw read data of the 23 selected genomic projects were downloaded from the Sequence Read Archive - SRA of the NCBI public database and assembled with SPAdes. If the isolate had assembly data available, the metrics of the two assemblies were compared, and the contig set that showed the better N50 value was kept for further analysis. Only six genomes of the studied had better N50 values than our assemblies. All of them were deposited at the CryptoDB database v43: *C. hominis*: UKH1, TU502_2012, 30976, 37999, UdeA01, and *C. meleagridis* UKMEL1 (Table S2). Only for three genomes, it was not possible to find a previous assembly data available: *C. hominis* (SWEH2 - SWEH5) and *C. meleagridis* (TU1867). To exclude possible contaminating contigs, BLASTN comparisons with a *Cryptosporidium* genome database (reference genomes of *C. hominis* UdeA01, *C. parvum* Iowa II and *C. tyzzeri* UGA55) were performed, and only those with Bit score value $\geq 300$ were kept for further analyses. Selected genomes assemblages' statistics are also shown in Table S2.

*Cryptosporidium* genome size is close to 9.0 Mb in most studied isolates; only three have genome assemblies below 9.0 Mb; 8.2 Mb, in *C. parvum* anthroponotic UKP14, and 8.8 Mb in both *C. hominis* SWEH2 and SWEH5 isolates. Four genomes had a mean read coverage below 50X, three of which belong to *C. hominis* species (UKH3, SWEH2, and SWEH5) and the *C. parvum* isolate UKP5. The most fragmented genomes were in the *C. parvum* species, isolate UKP3 and the anthroponotic UKP14 with 2,971 and 2,787 contigs, respectively, with coverage above 229X and 69X. All genomes have an average GC content of 30% and ambiguities that do not exceed 0.26% (Table S2).

### Single nucleotide variants detection

To identify single nucleotide variants -SNVs- throughout the genomes, inter and intra-species comparisons were made aligning the *de novo* assembled contigs of each isolate against the reference genome of *C. parvum* Iowa II. *Cryptosporidium parvum* isolates aligned more than 99.2% of the contig bases to the Iowa reference, except for the anthroponotic genomes, UKP14 and UKP15, that aligned 97%. *C. hominis* genome data behave similarly, with an aligned ratio that ranges between 99 to 99.38%, except for UKH4 that reached 98%. In *C. meleagridis*, a lower alignment rate was achieved with an overall rate close to 97% (Table S3).

The global nucleotide identity showed the expected results based on the phylogenetic relationship among the three species. While *C. parvum* genome identity ranged between 99.51% and 99.93% within the species, *C. hominis* genomes showed an identity of around 96.8% compared to the Iowa reference genome. *Cryptosporidium meleagridis* confirmed its higher distance with *C. parvum* with a global identity of around 91.5%.

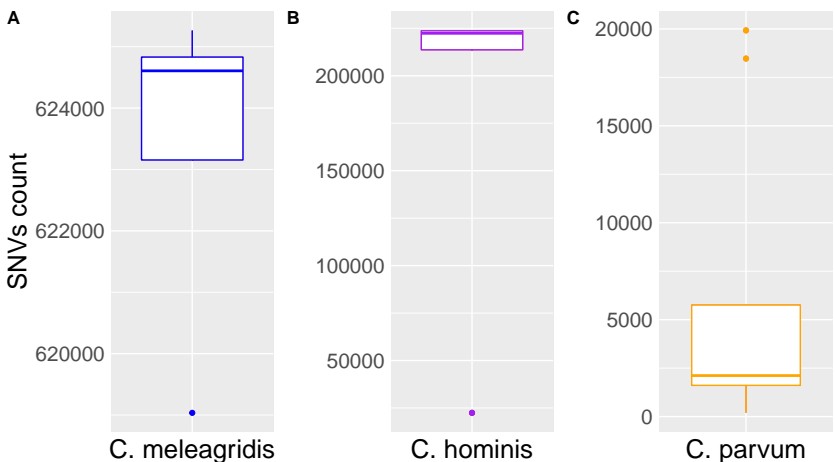

**Figure 1** **Accumulation of SNVs in Cryptosporidium species.** Box plots of the *Cryptosporidium* species analyzed and the number of total SNVs detected. The medians of the SNVs found in each species are (A) *C. meleagridis* (624,607), (B) *C. hominis* (223,640), and (C) *C. parvum* (2,116). Outliers in *C. meleagridis* and C. hominis correspond to the UKMEL3, SWEH2, and SWEH5, respectively. Concerning *C. parvum*, outliers correspond to anthroponotic UKP14 and UKP15 isolates.

Single nucleotide variants within the *C. parvum* genomes ranged between 1,595 and 5,752, except for the anthroponotic isolates that showed more than 18,000 SNVs. When *C. hominis* genomes were compared with the *C. parvum* Iowa II reference, around 220,000 SNVs were detected in each isolate, while in *C. meleagridis* genomes, the number when up to more than 600,000 (Fig. 1).

## Phylogenomic analysis

To verify the topology and taxonomic location described in the genus *Cryptosporidium*, the SNV data was used to generate a nucleotide matrix. Then, an unrooted maximum likelihood (ML) tree was constructed, including the 24 evaluated genomes (23 de novo assembled genomes and the reference genome Iowa II). The transversion model (TVM) was selected as the best model of evolution Table S4). The complete matrix comprised 800,861 sites and the best tree had had a likelihood value of −3401691.458. The tree obtained shows three monophyletic clades, following the actual classification scheme, that group the 24 genomes of the three *Cryptosporidium* species included in this work. The branch supports are optimal and have a 100% agreement between SH-aLRT and ultra-fast bootstrap for the main branches, with discrepancies in three internal *C. parvum* nodes and one internal *C. hominis* node (Fig. 2).

Within the *C. parvum* species, the presence of two separate branches with statistical supports was observed, which allowed the segregation of the anthroponotic isolates UKP14 and UKP15 in a separate branch of the zoonotic isolates with a 100% support. Furthermore, it is noteworthy to point out that all the *C. parvum* isolates that belong to the gp60 gene IIa subtype family grouped with a 100% bootstrap, including the isolate Iowa II. In the case of *C. hominis* clade, the isolates seem to segregate with statistical support according to its subtype family, except for isolate 30976. The *C. meleagridis* clade is also supported with

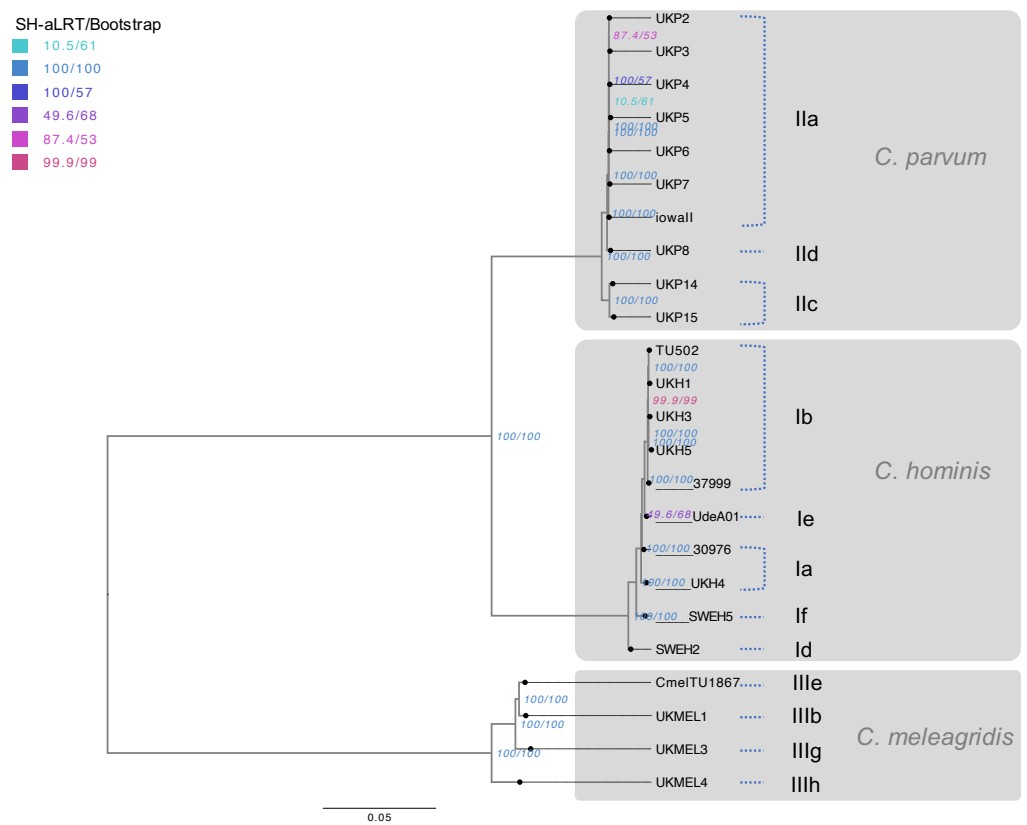

**Figure 2 Phylogenomic analysis of *Cryptosporidium* species.** Maximum Likelihood (ML) tree was based on the SNVs present within the 24 *Cryptosporidium* genomes studied. The presented tree is unrooted, and the longest branch was selected as an arbitrary outgroup. The horizontal scale line represents the number of base substitutions per site analyzed. TVM was the evolutionary model. The supports of the branches are based on an analysis of 1000 **aLRT** replications (%) / ultra-fast **bootstrap** replications (%). Red branches have an aLRT and bootstrap value of 100%. Branches are labeled with the isolate ID and allelic family based on the *gp60* gene.

100% bootstrap, and the phylogenetic signal is enough to separate the 4 different isolates according to its subtype family. In general, all genomes were grouped according to the species classification and its gp60 gene subtype family (Fig. 2).

## Single nucleotide variants in coding regions

SNVs located in coding regions were identified in all the genomes through the read mapping analysis against the *C. parvum* Iowa II version 43 of CryptoDB. As expected, compared with the *C. parvum* IOWA reference, the genomes of *C. hominis* and *C. meleagridis* showed the highest number of these variants in coding regions with more than 150,000 and 400,000, respectively, and nearly 60% corresponding to synonymous changes (sSNVs). There were no intraspecies differences in the accumulation of variants in coding regions or synonymous and non-synonymous changes in both species. On the contrary, in *C. parvum* isolates, the number of SNVs in CDSs was less than 20,000 in all genomes, with more than 50% corresponding to non-synonymous mutations. The results in *C. parvum* suggest an

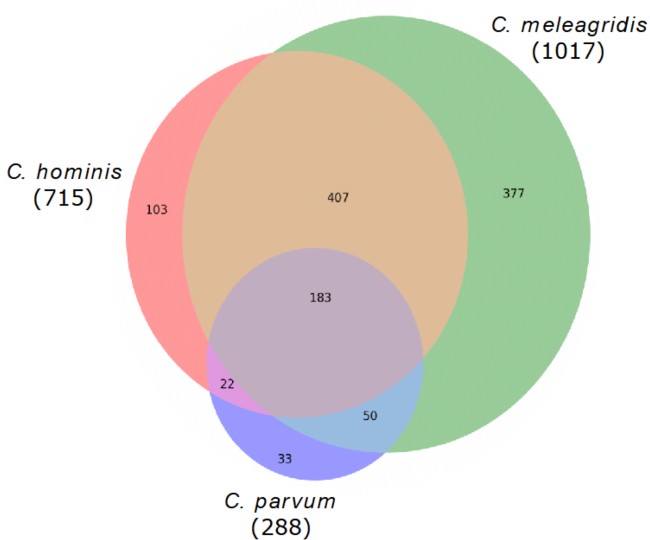

**Figure 3** **Genes detected with deleterious mutations in the three *Cryptosporidium* species.** Venn diagram showing the genes with shared and exclusive deleterious mutations in the three *Cryptosporidium* species included in the study.

intraspecies heterogeneity with values ranging between 840 and 3,476 SNVs in CDSs in zoonotic isolates and around 14,000 in anthroponotic isolates. However, these differences were not statistically significant. After characterizing the synonymous and non-synonymous mutations in coding regions, the genes with nsSNVs were analyzed separately, finding 2,532 in at least one genome of each species, with 110 exclusives for *C. meleagridis*, 11 for *C. hominis,* and 2 for *C. parvum.*

Genes with non-synonymous variants (nsSNVs) were annotated by the SIFT4G effect predictor, and those with non-tolerated changes and possibly associated with deleterious mutations were identified. With this strategy, a total of 1,017 genes were predicted with deleterious mutations in *C. meleagridis*, 715 in *C. hominis,* and 288 in *C. parvum.* Of these, 183 are shared by all three species, being present in at least one genome of each species; 377 are exclusive to *C. meleagridis*; 103 to *C. hominis,* and 33 to *C. parvum* (Fig. 3).

Only 29.5% of the genes annotated with deleterious mutations by SIFT, and shared by the three species, could be annotated with EggNOG mapper. Enrichment analysis in GO terms indicated that they are mainly involved in biological processes and metabolic processes with molecular functions such as catalytic or binding activity.

The functional orthology analysis performed on the KAAS server against the KEGG database had a better performance compared to the previous GO assignment, annotating 46.5% of the shared genes and indicating that most of them are involved in enzymatic processes, DNA repair, recombination, and proteins associated with the chromosomes (Fig. 4). Complementary annotation determined that only 1.1% of the genes encoded proteins secreted by the classical pathway, whereas 10.9% encoded for proteins secreted

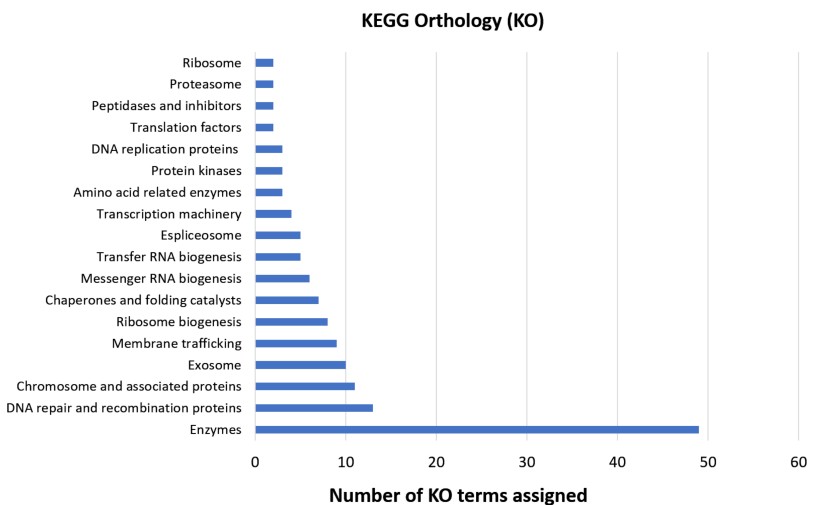

**Figure 4** KEGG orthology analysis of genes with deleterious mutations shared by the three *Cryptosporidium* species. Assignment of KO terms to genes with deleterious mutations detected in the three *Cryptosporidium* species through the KAAS server in the KEGG database.

by non-classical systems. Additionally, 2.3% of the putative encoded proteins exhibited domains with transmembrane helixes.

Regarding genes that carry deleterious mutations within each species (species-specific mutated genes), 29.1 and 28.9% were classified in at least one GO category for *C. hominis* and *C. meleagridis*, respectively. In *C. parvum*, GO terms were assigned only in 3 genes (9.09%), classified exclusively in the cellular component category. The functional orthology analysis again achieved a better result compared to the EggNOG-mapper software, assigning 42.4% in *C. parvum* (Fig. 5A), 60.2% KO in *C. hominis* (Fig. 5B), and 59.1% in *C. meleagridis* (Fig. 5C). The assignment of functional orthologs in metabolic pathways established that in the three species, most of these genes are involved in enzymatic processes. The second most frequently assigned pathway in genes with species-exclusive deleterious mutations was ribosome transfer RNA biogenesis in *C. parvum*, spliceosome complex in *C. hominis*, and biogenesis in *C. meleagridis*.

## Indel events analysis

Inter and intraspecies indel events were identified in the genomes through their comparison to the *C. parvum* Iowa II reference genome. For simplicity reasons, insertions and deletions were referred, assuming as reference the *C. parvum* Iowa II genome. Observed insertions were in the range of 50 to 500 nucleotides. In *C. parvum*, the highest number of insertions was detected in the zoonotic isolates UKP3 and UKP4. No statistically significant differences were found in the accumulation of insertions among the studied species. Deletions identified in the genomes had a similar size to that described for the insertions, with the majority falling into the range of 50 to 500 nucleotides. The highest number was observed in the genomes of *C. meleagridis*, followed by *C. hominis*. However, no significant differences were found in the accumulation of deletions among species.

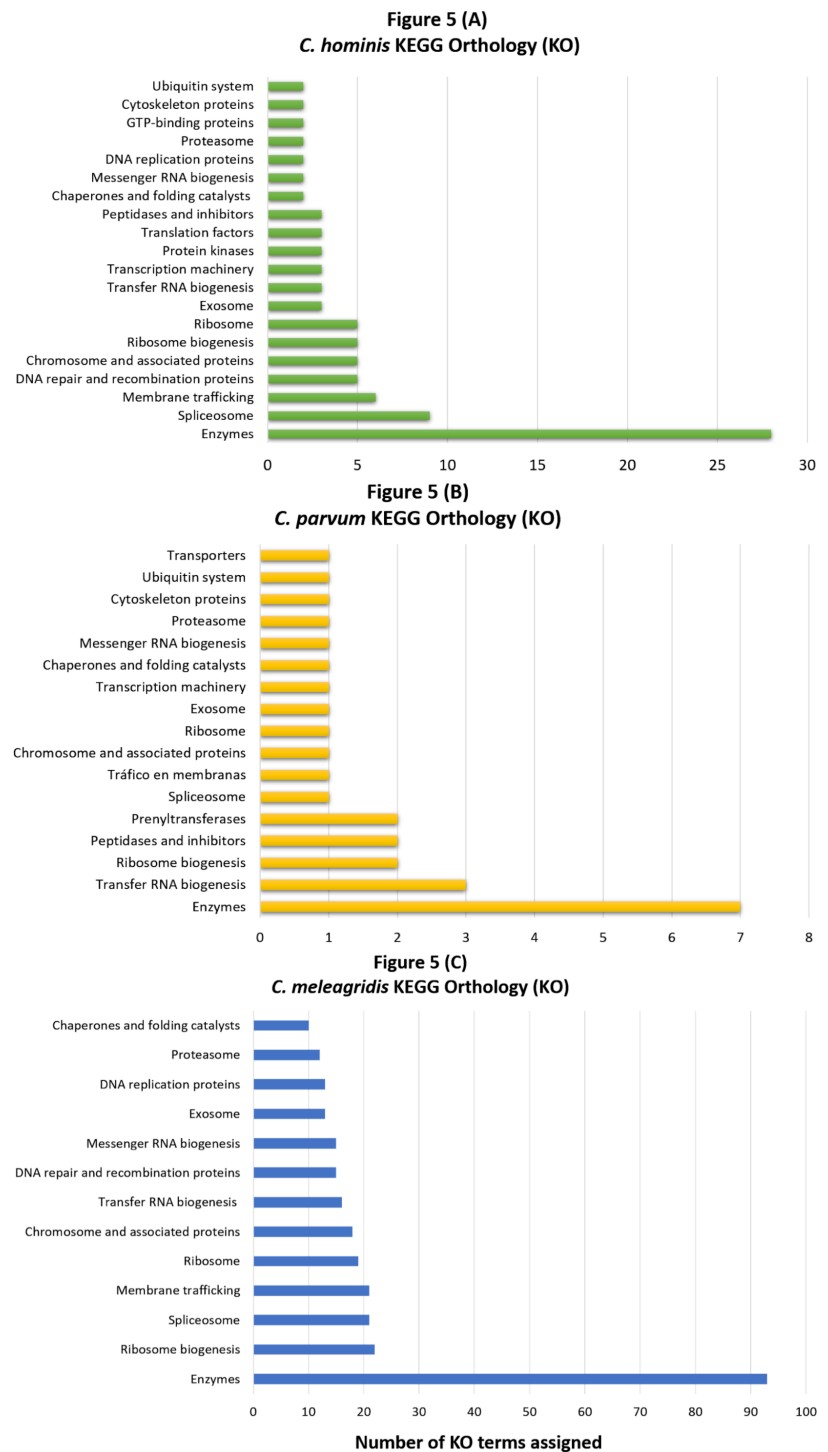

**Figure 5  KEGG orthology analysis of genes with species-exclusive deleterious mutations.** Assignment of KO terms to genes with species-exclusive deleterious mutations via the KAAS server in the KEGG database in *C. hominis* (A), *C. parvum* (B), and *C. meleagridis* (C).

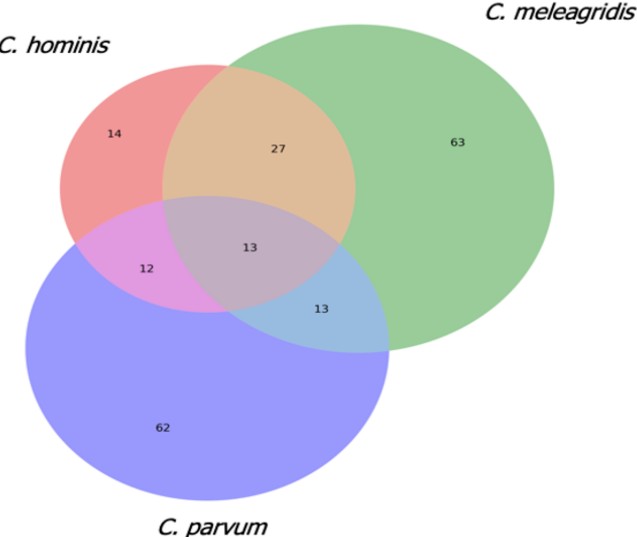

**Figure 6** **Analysis of genes with indel events in the three *Cryptosporidium* species.** Venn diagram that shows the genes with indel events that are shared and exclusive in the three *Cryptosporidium* species.

Coding genes that presented indels in the three species were identified and annotated, whenever possible. We found 322, 215, and 176 genes with indels in *C. hominis*, *C. parvum*, and *C. meleagridis*, respectively, but only 13 were common to all 23 *Cryptosporidium* isolates. In both *C. meleagridis* and *C. hominis*, deletions were the most frequent indel event, corresponding to 76.1% and 59% of all structural variants, respectively. The *C. hominis* isolates UKH4 was the exception, in which insertions predominated. In contrast, in *C. parvum*, deletions were less common, representing 38% of the structural variants detected in the CDSs.

To gain insights into the gene losses occurring in the 23 isolates, those genes with indel events present in at least two genomes/isolates in each species were characterized. A total of 116 genes were selected in *C. meleagridis*, 100 in *C. parvum*, and 66 in *C. hominis*. Sixty-three were exclusive to *C. meleagridis*, 62 to *C. parvum*, and 14 to *C. hominis* (Fig. 6).

According to the annotation deposited in CryptoDB Version 46, of the 13 genes with insertion and deletion events shared by the three species, 40% code for uncharacterized proteins and the remaining for hydrolases (cgd8_1220), proteins with recognition motifs of RNA (cgd3_4150), mucins (cgd7_4020), among others. Twenty-seven genes with indels were identified in all *C. meleagridis* genomes, with 56% of them corresponding to uncharacterized proteins. Regarding *C. parvum*, insertions in the genes cgd3_190 (involved in the formation of fibrillin) and cgd3_3900 (uncharacterized protein) were present in 56% of the evaluated genomes. An interesting finding was that all *C. hominis* genomes exhibited indels in three genes (orthologs of cgd6_4290, cgd7_420, and cgd7_500); with cgd6_4290 and cgd7_420 being also affected in all genomes of *C. meleagridis* (Table S5).

# DISCUSSION

Advances in DNA sequencing technologies and bioinformatics have promoted the routine use of complete genome sequences, revolutionizing the study of both model and non-model organisms, particularly in microbiology (*Young & Gillung, 2020*). Phylogenomic is one of the numerous disciplines that have taken advantage of the progress in NGS technologies, using the massive datasets to infer both phylogenetic relationships between taxa, improving the understanding of molecular evolution, and putative functions for DNA or protein sequences (*Young & Gillung, 2020*).

A phylogenomic analysis is being applied now to validate previous findings with classic single-marker molecular methods and to resolve with unprecedented resolution, the phylogenetic relationships between species and isolates of the human infecting *Cryptosporidium* parasites (*Glaberman et al., 2001*; *Abe & Makino, 2010*; *Abal-Fabeiro et al., 2013*; *Wagnerová et al., 2015*; *Pérez-Cordón et al., 2016*). However, this phylogenomic approach has focused on *C. hominis* and *C. parvum* by studying multiple concatenated loci (*Feng et al., 2017*; *Nader et al., 2019*), or different variable positions that represent below 1% of the genome (*Gilchrist et al., 2018*). To gain insights into the evolutionary relationships in *Cryptosporidium,* a comparative and phylogenomic study was conducted with 23 genomes of the most frequent species in humans: *C. hominis*, *C. parvum*, and *C. meleagridis.*

The sequence identity between *C. hominis* genomes (96.85%) and the reference genome of *C. parvum* Iowa II found in our study confirms the high similarity between both species and agrees with previous findings that report differences of a maximum of 3% (*Xu et al., 2004*; *Mazurie et al., 2013b*; *Zahedi et al., 2013*). These studies describe that both genomes exhibit remarkable structural conservation, and some authors suggest that the phenotypic differences may be due to subtle variations in the sequences of genes that code for the interface proteins between the parasite and its host (*Xu et al., 2004*; *Mazurie et al., 2013b*; *Zahedi et al., 2013*). Regarding *C. parvum*, there was a lower percentage of aligned blocks against the reference genome in the anthroponotic isolates compared to the zoonotic ones, which agrees with other studies that have also described differences in the genomic structure and nucleotide content between zoonotic and anthroponotic isolates of *C. parvum* (*Widmer et al., 2012*; *Fan, Feng & Xiao, 2019*; *Nader et al., 2019*). As it was expected, *C. meleagridis* genomes had a lower global identity against *C. parvum* reference genome than that detected with *C. hominis*, which agrees with the phylogenetic relationships described for the three species using different genetic locus (*Šlapeta, 2013*; *Khan, Shaik & Grigg, 2017*). This finding may be related to the differences in host specificity reported in the three species, where *C. hominis* and *C. parvum* have host ranges limited to mammals, while *C. meleagridis* is described in mammals and birds (*Šlapeta, 2013*; *Khan, Shaik & Grigg, 2017*). It has been suggested that mammals were possibly the original hosts for *C. meleagridis* and that later this species adapted to birds (*Caccio & Widmer, 2014*).

Single nucleotide variants (SNVs) results obtained in this study showed similar behavior in their number and frequency in the genomes of *C. hominis* with around 220,000 variants, compared to the *C. parvum* IOWA reference, results that differ from those reported in

previous comparative studies (*Isaza et al., 2015*; *Gilchrist et al., 2018*). Isaza et al. (*Isaza et al., 2015*) identified an average of 43,258 SNVs between the reference genome *C. parvum* Iowa II and the *C. hominis* UdeA01, UKH1, and TU502_2012 isolates (deposited in CryptoDB version 8). One possible explanation of the lower number of SNVs detected in this work could be the different methodological approaches used to identify these variants. Authors only considered those SNVs located at 30 nucleotides or more from regions where the alignment failed, underestimating the total number of SNVs. Additionally, the genomes used in Isaza's study were obtained from the CryptoDB version 8 database, whose genomes had a lower degree of purification than those used in our work (version 43). *Gilchrist et al. (2018)* also found a lower number of SNVs (36,780) in a comparative study of 32 genomes of *C. hominis*. The authors mapped the short reads of the *C. hominis* genomes against the genome of *C. parvum* Iowa, which was sequenced with Pacific Biosciences (PacBio). It could be possible that the assembly of a genome through long sequencing reads can affect the number of variants detected. Although it is not clear how long reads sequencing methods might differ from comparative genomic approaches using short-read data, i.e., Illumina (*DeMaio et al., 2019*), it has been shown that error rates on this platform are much higher than those recorded for Illumina (15% versus 0.1%) and usually is performed at a lower read coverage compare to Illumina projects.

Concerning the single nucleotide variants among the *C. parvum* genomes evaluated in this study, a more significant number of SNVs was identified in the anthroponotic isolates with more than 18,000 compared to the zoonotic isolates, which ranged between 1,595 and 5,752 SNVs. These results are similar to the findings reported by *Widmer et al. (2012)*, who identified about 16,606 SNVs by comparing the genomes of two *C. parvum* isolates with different host ranges, the anthroponotic *C. parvum* TU114 and the reference genome of the zoonotic Iowa II isolate. The differential accumulation of intragenotypic SNVs in *C. parvum* reflects the genetic diversity of this species, which is possibly related to the evolution of the parasite influenced by the selective pressures in both humans and animal hosts. For this reason, it has been postulated that the accumulation of genomic variants could influence the host range (*Weir et al., 2011*; *Blake et al., 2015*; *Grinberg & Widmer, 2016*). Concerning *C. meleagridis*, more than 600,000 SNVs were identified in the genomes compared against the reference genome of *C. parvum*. It is essential to highlight that there are currently no reports on the structural variations like SNVs in this species, being this the first one.

In this work, a comprehensive phylogenomic analysis using more than 800.000 single nucleotide variations detected in 24 genomes of the species *C. parvum*, *C. hominis*, and *C. meleagridis* was done. To our knowledge, this is the most extensive phylogenomic analysis carried out within the genus and one of the largest within the Phylum Apicomplexa. One of the main findings in this study is that, although we didn't select neutral evolving positions within the genomes, there was a strong phylogenetic signal, supported by two statistical tests, that allowed the well-supported segregation for most of the isolates. However, discrepancies were found in three internal *C. parvum* nodes and one in *C. hominis*. It is essential to highlight that the mentioned differences did not alter the global topology of the phylogenomic tree obtained. In related literature, it has been described that the mismatches

in the supports of the branches obtained by bootstrapping and probability ratio tests can arise as a consequence of performing the analysis with small samples and with highly heterogeneous nucleotide substitution models (*Guindon et al., 2010*).

The obtained tree confirms the previously reported topology for these three intestinal species of *Cryptosporidium*, inferred from single-locus phylogenetic studies, and by the use of different loci such as 18S rRNA, gp60, and other polymorphic genes (*Ren et al., 2012*; *Šlapeta, 2013*; *Nader et al., 2019*). *Cryptosporidium meleagridis* is confirmed as the most divergent group among the three studied species (*Šlapeta, 2013*; *Khan, Shaik & Grigg, 2017*). At the species rank, the bootstrap and SH-aLRT statistical support was 100%. Furthermore, most of the isolates were grouped according to its *gp60* gene family type, regardless of its geographical origin.

The phylogenomic analysis of the *C. parvum* isolates evidenced the separation of its central clade into two branches with significant statistical support (100%), with zoonotic isolates in one branch and anthroponotic isolates in the other. This finding agrees with that reported by several authors (*Widmer et al., 2012*; *Danišová et al., 2017*; *Feng et al., 2017*; *Nader et al., 2019*), in which through unilocus phylogenetic analyzes and multiple concatenated loci between anthroponotic and zoonotic isolates, determined that the anthroponotic isolates of *C. parvum* formed a separate group from the zoonotic isolates. *Nader et al. (2019)*, through phylogenetic analysis of neutrally evolving coding loci across 21 *Cryptosporidium* isolates, identified two *C. parvum* lineages with distinct host-specificity, which were designated as *Cryptosporidium parvum parvum* ( zoonotic) and *C. p. anthroponosum* (anthroponotic). Additionally, they found that human infective *C. hominis* and *C. parvum* isolates form a distinct superclade along with *C. cuniculus*, another species associated with human infections. Subsequent analysis of high-quality SNPs detected in 16 genomes of the two *Cryptosporidium parvum* subclades, confirming the zoonotic *C. p. parvum* and anthroponotic *C. p. anthroponosum* subspecies designation (*Nader et al., 2019*).

*Feng et al. (2017)* evaluated the SNVs accumulation in several genomes of *C. parvum* IIa and IId families, which preferentially infect calves and lambs, respectively, in some European countries. This study revealed that most of the SNVs occur in subtelomeric regions of the chromosomes, with a high percentage located in coding regions of the genome, and near the half being non-synonymous. Additionally, the subtypes evaluated shared more than 50% of SNVs, and phylogenetic analysis of the SNVs data showed a robust separation of IIa sequences and IId sequences, and a high divergence with reference Iowa genome (*Feng et al., 2017*). These findings agree with the results obtained in this study because SNVs show a concordant relationship with gp60 subfamily typing.

Conversely, *Gilchrist et al. (2018)* studied the genetic diversity of thirty-two genomes of different *C. hominis* subtypes isolated from children with poor living conditions from Bangladesh. They found 36,780 SNVs that varied between the *C. hominis* isolates, with a homogeneous distribution throughout the genome and only 4% occurring with a frequency greater than 20%. These authors also built a phylogenetic tree based on the SNVs (1,582) found in those genomes, in which no groupings regarding the subtype family was observed, concluding that the use of a single marker (gp60) does not reflect the evolutionary changes
of the entire genome and, in turn, confirming the weakness of the typing of unique markers for taxonomic assignments within this genus. Our findings reinforce this argument since an analysis with a more significant number of positions of different isolates considerably improves the resolution power compared to that obtained from unilocus analysis or multiple concatenated loci or partial fragments of the genome. Another aspect that could influence the topology of the phylogenomic tree obtained by Gilchrist et al. is the high rate of recombination on chromosome 6 reported among circulating isolates from endemic countries for *C. hominis* (*Li et al., 2013*; *Zahedi et al., 2013*). This feature is associated with greater genetic variability and the generation of hypertransmittable subtypes and favors a wide distribution of gp60-based allelic families in the phylogenetic tree without a cluster aggregation. The recombination phenomenon was not a relevant variable in our study since genomes analyzed correspond to isolates circulating in different geographic locations of four continents.

Several authors have proposed that phenotypic differences between *Cryptosporidium* species are related to polymorphisms on protein-coding regions (*Xu et al., 2004*; *Pain, Crossman & Parkhill, 2005*; *Bouzid et al., 2013*; *Nader et al., 2019*). In our study, the single nucleotide variants located in CDSs were analyzed and characterized as synonymous and non-synonymous changes. Compared to the *C. parvum* reference, *Cryptosporidium* species that showed the highest number of SNVs in protein-coding regions were *C. meleagridis* and *C. hominis* than 400,000 and 150,000 variants, respectively. Forty-two percent of them corresponded to non-synonymous changes. Our results on the number of SNVs in *C. hominis* differs from the data reported previously, and similarly, the same occurs with SNVs in coding regions. *Isaza et al. (2015)* identified 36,753 SNVs located on coding regions in the genome of four isolates of *C. hominis* (version 8 of CryptoDB), using as a reference the genome of *C. parvum* Iowa II. As we mentioned before, these discrepancies may be related to the methodological approach used in every study.

In *C. parvum*, the number of variants in coding regions was below 20,000 SNVs, with intraspecies differences related to a heterogeneous behavior between zoonotic and anthroponotic isolates. Additionally, it was identified that more than 50% of the SNVs located in coding regions correspond to non-synonymous mutations. In a previous study carried out by *Widmer et al. (2012)*, non-synonymous SNVs were present in a range from 28% to 32% of all SNVs, and 60% of all nucleotide positions in the two genomes were not synonymous.

Analysis of the deleterious mutations in the *Cryptosporidium* species evaluated in the study showed that 183 genes were predicted with mutations that were present in at least one genome of each species: 377 exclusive to *C. meleagridis*, 103 to *C. hominis,* and 33 to *C. parvum*. Unfortunately, 53,5% of the genes with deleterious changes were located in uncharacterized coding regions (hypothetical proteins), so the biological impact of these mutations could not be determined. Annotation of the shared and species-exclusive genes with deleterious mutations establish that most encode enzymes and proteins involved in DNA repair, recombination processes, proteins associated with the chromosomes, as well as the biogenesis of transfer RNA and ribosomes. Comparative analyzes carried out previously have found that the genes with the highest number of SNVs in *C. hominis* and

*C. parvum* were related to ribosome assembly, translation processes, and coding genes for proteins with transmembrane domains (*Widmer et al., 2012*; *Isaza et al., 2015*). *Sikora et al. (2017b)* carried out an intraspecies comparative analysis of 14 genomes of *C. hominis*. They found 18 genes with non-synonymous mutations, with only the gene that codes for a protein of the oocyst wall COWP9 (oocyst of cgd6_210) with a deleterious mutation, which was present in eight of the fourteen genomes. Mutated genes annotated in our study also code for surface proteins with transmembrane domains and proteins secreted by non-classical pathways, suggesting that the interaction processes between the parasite and the host cells could be affected. This finding agrees with that reported by other authors who have described that the processes that are mainly affected are adhesion and invasion (*Li et al., 2013*; *Isaza et al., 2015*; *Gilchrist et al., 2018*; *Widmer, 2018*; *Xu et al., 2019*). It has been determined that the proteins secreted by non-classical pathways usually are growth factors, inflammatory cytokines, components of the extracellular matrix that regulate cell differentiation, proliferation, and apoptosis, as well as surface proteins in parasites involved in the initial interaction with the host (*Nickel, 2003*). This reinforces the need to improve the annotation of *Cryptosporidium* genomes, allowing the understanding of unknown aspects related to evolution, virulence, and pathogenicity in this genus.

Indels events in the genomes of the 23 *Cryptosporidium* isolates were also evaluated in our study. More than 60% of these variants were located in CDSs, a finding that could be expected since the *Cryptosporidium* genome has a percentage of coding regions more significant than 70% (*Nader et al., 2019*; *Xu et al., 2019*). The highest number of deletions, using reference *C. parvum* IOWA, occurred in *C. meleagridis* genomes, followed by *C. hominis*, suggesting a partial loss of genome fragments these species. Indel events were less abundant than SNVs in the 23 genomes analyzed, contrary to the reports in other apicomplexan such as *Plasmodium falciparum*, in which they have been described as the dominant mechanism of polymorphism within the genome (*Miles et al., 2016*). *Feng et al. (2017)* identified 1,200 insertion events and 1,500 deletions in a comparative genomic study of *C. parvum* isolates. In our study, which analyzed a more extensive number of genomes, we found less than 100 structural variations in each of the zoonotic and anthroponotic isolates. Previous studies have determined that indels are significantly more frequent in the peri-telomeric and subtelomeric regions of *Cryptosporidium* genomes (*Nader et al., 2019*). *Guo et al. (2015)* analyzed five genomes of *C. hominis* against the reference *C. parvum* Iowa, and they identified several insertions and deletions near the telomeres on chromosome 6, associated with recombination events, which could indicate that the duplication or deletion of subtelomeric genes is involved in the differences in host specificity between *Cryptosporidium* species. These recombination events can also explain the low support obtained in the phylogeny for the *C. hominis* 30976 isolate. Members of multicopy gene families and under a strong positive selection, such as MEDLE proteins, insulin-like proteases, and mucin-type glycoproteins, related to the parasite-host interaction, are ubicated in these regions (*Fan, Feng & Xiao, 2019*; *Feng & Xiao, 2019*; *Xu et al., 2019*).

Although several studies have described deletions in genes encoding the MEDLE proteins in *C. parvum* and *C. hominis* (*Widmer et al., 2012*; *Liu et al., 2016*), in the present study, no structural variants were found in these genes or those encoding insulin-like proteases.

However, deletions were found in all genomes of *C. meleagridis* and at least two isolates of *C. hominis* and *C. parvum* in a gene coding for a cryptosporidial mucin (ortholog of cgd7_4020), also known as gp900. This is a microneme secreted surface glycoprotein encoded by a single copy gene; it is involved in the apical portion of sporozoites and merozoites to enterocytes, which is required to initiate the invasion process (*Okhuysen & Chappell, 2002*; *Carruthers & Tomley, 2008*). However, since there is a repertoire of adhesion proteins in *Cryptosporidium*, including the proteins related to thrombospondin, p23, the gp40 / p30 protein complex, and the Circumsporozoite-like protein—CSL (*Langer-Curry & Riggs, 1999*; *Bouzid et al., 2013*), the alterations identified in the gp900 gene probably do not affect the interaction processes between the invasive stages of the parasite and the host cell. Additional analyzes are required to determine the biological implications of these deletions in the binding and invasion process, mainly in isolates of *C. meleagridis*.

Another interesting finding in this study was identifying deletions in all genomes of *C. meleagridis* and *C. hominis* for genes that encode for proteins with a WD-40 (cgd6_4290 ortholog) and SNF2/DEXDc/HELICc domains (cdg7_420 ortholog). WD-40 domain has tryptophan-aspartic acid (WD) repeats of approximately 40 amino acids and is considered one of the ten most abundant protein domains in eukaryotes (*Xu & Min, 2011*; *Jain & Pandey, 2018*). The proteins that contain these repeats are involved in various cellular processes, acting as an adaptor in many different protein complexes or protein-DNA complexes, signal transduction, transcription, cell cycle regulation, and apoptosis; however, no enzymatic activity has been assigned (*Stirnimann et al., 2010*; *Xu & Min, 2011*). These domains have been reported as highly polymorphic in other apicomplexan protozoa, such as *Plasmodium falciparum*, suggesting the participation of WD-40 in basic cellular and metabolic processes (*Chahar et al., 2015*). SWI2/SNF2 (Switching defective -SWI and Sucrose nonfermenting-SNF) protein family are ATP-dependent chromatin remodeling factors that modulate the access of transcription factors to regulatory regions of genes (*Sullivan et al., 2013*). Previous reports indicate homologs of these domains in different apicomplexan, including *Plasmodium falciparum* (*Ji & Arnot, 1997*) and *Toxoplasma gondii* (*Sullivan et al., 2003*). *Cryptosporidium* parvum has 14 chromatin-remodeling SNF2/SWI2 ATPases (*Templeton et al., 2004*). Alterations in the genes encoding these proteins could affect the epigenetic regulatory mechanisms in this genus.

Exclusive insertions were identified in 50% of the *C. parvum* isolates in the cgd3_190 gene. This gene encodes a microneme secreted protein with epidermal growth factor - EGF domains like, which are involved in cell signaling (*Carruthers & Tomley, 2008*). In other apicomplexans, such as *Toxoplasma gondii*, these domains have been associated with adhesion processes to the host cell (*Huynh, Boulanger & Carruthers, 2014*). It has also been shown that in the presence of calcium, the EGF domains adopt an extended structure resistant to proteases, favoring the interaction of the N-terminal portion of the molecule with the host cell ligands, favoring invasion (*Carruthers & Tomley, 2008*).

## CONCLUSIONS

Here we present, to our knowledge, the most comprehensive phylogenomic and genomic comparative analysis performed in the most relevant human infecting *Cryptosporidium*

species, which includes complete genomes from different isolates, allelic families, and subtypes. Comparative analysis of more than 800,000 single nucleotide variable positions detected in 24 genomes of the three main species infecting humans (*C. parvum, C. hominis,* and *C. meleagridis*) generate a more robust analysis on the phylogenetic relationships between the *Cryptosporidium* species of human public health concern. This phylogenomic analysis also confirmed the *gp60* loci segregation pattern observed in subtype families. Most of the SNVs and indels detected in the study genomes were ubicated in coding regions. Genes with deleterious changes and indels were identified and annotated, whenever possible, in the three species. These mutated genes were associated with the processing of genetic information and enzymatic and metabolic processes; however, most of them remain uncharacterized and encode hypothetical proteins.

### Funding

This work was funded by the COLCIENCIAS Convocatoria 777-2017 Para Proyectos de Ciencia, Tecnología e Innovación en Salud 2017, project name: "Estudio de prevalencia, diversidad genética y genómica de Cryptosporidium en población VIH positiva de Antioquia" code 115-777 57608, and Vicerrectoría de Investigación (CODI), Universidad de Antioquia, grant "Finalización del genoma de referencia de C. hominis aislado UdeA01", code 2017-16-171. The funders had no role in study design, data collection and analysis, decision to publish, or preparation of the manuscript.

### Grant Disclosures

The following grant information was disclosed by the authors:
COLCIENCIAS Convocatoria 777-2017 Para Proyectos de Ciencia, Tecnología e Innovación en Salud 2017, project name: "Estudio de prevalencia, diversidad genética y genómica de Cryptosporidium en población VIH positiva de Antioquia": 115-777 57608.
Vicerrectoría de Investigación (CODI), Universidad de Antioquia, grant "Finalización del genoma de referencia de C. hominis aislado UdeA01": 2017-16-171.

### Competing Interests

The authors declare there are no competing interests.

### Author Contributions

- Laura M. Arias-Agudelo performed the experiments, analyzed the data, prepared figures and/or tables, authored or reviewed drafts of the paper, and approved the final draft.
- Gisela Garcia-Montoya and Ana L Galvan-Diaz performed the experiments, authored or reviewed drafts of the paper, and approved the final draft.
- Felipe Cabarcas performed the experiments, analyzed the data, authored or reviewed drafts of the paper, and approved the final draft.
- Juan F. Alzate conceived and designed the experiments, performed the experiments, analyzed the data, prepared figures and/or tables, and approved the final draft.

## Data Availability

The Cryptdb Assemblies datasets, ChominisUKH1, Chominis37999, ChominisTU502, Chominis30976, ChominisUdeA01, and CmeleagridisUKMEL1, are available at CryptoDB (Release-43 Download).

SRA NGS reads

Isolate ID BioProject ID SRA

UKH1 PRJNA222837 SRR1015721
UKH3 PRJNA253834 SRR6131684
UKH4 PRJNA253838 SRR6143718
UKH5 PRJNA253839 SRR6144056
37999 PRJNA252787 SRR1558150
TU502_2012 PRJNA222836 SRR1015747
30976 PRJNA252787 SRR1557959
UdeA01 PRJEB10000 ERX1047563
SWEH2 PRJNA307563 SRR3098103
SWEH5 PRJNA307563 SRR3098109
UKP2 PRJNA253836 SRR6117460
UKP3 PRJNA253840 SRR6147472
UKP4 PRJNA253843 SRR6147581
UKP5 PRJNA253845 SRR6147587
UKP6 PRJNA253846 SRR6147945
UKP7 PRJNA253847 SRR6147964
UKP8 PRJNA253848 SRR6148259
UKP14 PRJNA315506 SRR6813718
UKP15 PRJNA315507 SRR6813719
UKMEL1 PRJNA222838 SRR1179185
UKMEL3 PRJNA315502 SRR6813720
UKMEL4 PRJNA315503 SRR6813896
TU1867 PRJNA192428 SRR793561

## Supplemental Information

Supplemental information for this article can be found online at http://dx.doi.org/10.7717/peerj.10478#supplemental-information.

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
