# Peer review of "Comparative genomic analysis of the principal Cryptosporidium species that infect humans"

_PeerJ, doi:10.7717/peerj.10478_

## Round 0.1 · original submission · Major Revisions

The review process is now complete, and two thorough reviews from highly qualified referees are included at the bottom of this letter. Although there is considerable merit in your paper, we identified some concerns that must be considered in your resubmission. I recommend the authors to review the phylogenetic analysis accordingly.

·

Basic reporting

See the general comments section

Experimental design

See the general comments section

Validity of the findings

See the general comments section

Additional comments

The report describes the results of the phylogenetic studies of Three Cryptosporidium species based on genome-wide SNV analysis. The report is straightforward and well prepared. I have mostly minor issues for the authors to consider.

Major comments:
1. L224-227 and Fig 1. The reason for the high SNVs within C. hominis and C. meleagridis is not clear. Was the C. parvum IOWA genome used as reference? This this is the case, the data are really not on intraspecies diversity as the descriptions implied.
2. The reason for different clustering by C. hominis isolate 30976 should be discussed in the context of genetic recombination. See: Guo, Y., Tang, K., Rowe, L.A., Li, N., Roellig, D.M., Knipe, K., Frace, M., Yang, C., Feng, Y., Xiao, L., 2015. Comparative genomic analysis reveals occurrence of genetic recombination in virulent Cryptosporidium hominis subtypes and telomeric gene duplications in Cryptosporidium parvum. BMC genomics 16, 320.

Other suggestions:
L22, L266. No need to capitalize the word Apicomplexan.
L24. Delete the word The.
L47. A parenthesis symbol is missing before “Furthermore”.
L137-138. The reads were extended. What does it mean?
L187, L209, L229. These subtitles are for the Results section. Please modify them.
L290. Delete the word coding.
Table 2. What is the differences between the last two columns of the table, which are both labeled as “Mean coverage of read mapping (X) “.

Reviewer 2 ·

Basic reporting

The premise for this work was that trees based on a few genes can lead to conflicts and the authors proposed to overcome this by conducting a phylogenetic analysis based on whole-genome seq variation. The authors downloaded 23 high-throughput short read datasets from SRA, from 3 Cryptosporidium species (10 for C. hominis, 9 for C. parvum and 4 for C. meleagridis), reconstructed assemblies and identified seq variants from the assembly alignments. A max likelihood phylogenetic analysis was conducted on the concatenated variant positions, which the authors say agrees with the accepted phylogenetic history for the three species and supported the monophyly of alleles from a locus, gp60, often used to type Cryptosporidium samples. This manuscript reflects a good amount of work, but there are serious flaws with the study and some opportunities missed as well, which maybe can be used as a basis for a solid comparative genomics work.

1. Basic reporting

The premise of the work --the need to use genome-wide data to resolve potentially conflicting gene-level phylogenetic incongruency (mentioned in the abstract only)—is flawed in this study: a) there is no mention in the intro to studies that have shown inconsistency between gene and species tree involving this three taxa, (b) two of the species used are very closely related and the third is a reasonably distant relative of theirs, so with this phylogenetic pattern incongruencies between gene-species tree will be exceedingly rare, (3) the authors use only three species (no outgroup) so there is an unrooted network of three taxa not a phylogeny, and finally (d) no reference is made to the premise in the introduction despite an extensive literature on the topic of gene tree-species tree conflicts. That is prob because this premise is a bit of a strawman, in that gene-species tree issues are not a problem for three taxa with this pattern of relationships.

The phylogenetic relationship between these three taxa should be discussed in the intro, as well as the extent of nucl sequence divergence between them (all of which are well established).

English is acceptable for the most part, accession numbers for the raw data are provided and the assemblies have been deposited with CryptoDB, the std, NIH-supported, public repository for these taxa. Figures need to use larger font (especially bootstrap values).

Experimental design

I applaud the authors for the effort of building genome assemblies and calling SNVs based on assembly alignment. Especially for the analysis of rapidly evolving genes, this approach to calling seq variants can be vastly more fruitful than read mapping, where reads may not map at all when there is extensive sequence divergence between the taxa being compared. This is where I thought they were going with this study, ie, determining if the comparison of assemblies countenances them to probe the evolution of genes (or segments of genes, or intergenic regions) that are not amenable to study from read mapping due to extensive seq divergence (or the presence to read mapping inconsistencies).

Another interesting topic that can be explored from assemblies (namely with several assemblies per species) that cannot be addressed through read mapping is whether there are genes conserved and specific to C. hominis or C. meleagridis that are absent in C. parvum (the source of the reference genome to which all data was aligned).

There are several problems with the phylogenetic analysis:
- expand on the phylogenetic methods. Namely, the name of the evolution model should be spelled out entirely the first time around and the abbreviation used after that. Same thing for the phylogenetic method and the programs used to assess support for internal branches. Models (Transversion Model, TVM) and methods should ALL be cited (some are but not all).
- Present table with BIC values that support the selection of TVM;
- The analysis uses no outgroup, so the phylogeny is unrooted (if rooting on longest branch, make sure to state);
- Most importantly: Fig 2 (phylogeny) is depicted with a scale, suggesting that branch lengths are proportional to change (also stated in the figure legend) but this not the case. What is depicted is an unscaled cladogram, with branches lengths proportional to number of branches. The (scaled) phylogram will show VERY long branches between species and nearly non-existent branches within species, since the intraspecies variation is miniscule (~2000 SNPs) compared to between species (200K – 600K variants).
- The authors state that some of the branching patters within species are not well supported. This is expected given that these are recombining eukaryotes, ie, no single tree is expected to represent the evolutionary history of the entire genome. This should be addressed.

Validity of the findings

The only aspect of the study that is novel is the generation of the assemblies. The raw data was generated by others, the relationship between species is well established, and no incongruency is expected between gene trees and species tree for this triad of taxa, so no phylogeny is required to use the complete genome. There is the potential that the gp60-based phylogeny is incongruent with the history of the whole genome (which, in a recombining organism, is an average of the history across all positions) but recombination can easily explain that. Again , the authors did not observe such incongrency, possibly because they only have 4-10 genomes from each species, with only 1 or 2 sequenced per subtype which limits the possibility of finding such inconsistencies…

Additional comments

There are interesting questions that can be addressed with the whole genome assemblies and it would be interesting to see what the authors come up with.

---

## Round 0.2 · accepted · Accept

The authors have satisfactorily responded to all questions raised by the reviewers and made the necessary changes to the manuscript.